# Molecular Cloning, Heterologous Expression, Purification, and Evaluation of Protein–Ligand Interactions of CYP51 of *Candida krusei* Azole-Resistant Fungal Strain

**DOI:** 10.3390/biomedicines11112873

**Published:** 2023-10-24

**Authors:** Tatsiana V. Tsybruk, Leonid A. Kaluzhskiy, Yuri V. Mezentsev, Tatyana N. Makarieva, Kseniya M. Tabakmaher, Natalia V. Ivanchina, Pavel S. Dmitrenok, Alexander V. Baranovsky, Andrei A. Gilep, Alexis S. Ivanov

**Affiliations:** 1Institute of Bioorganic Chemistry NASB, 5 Building 2, V.F. Kuprevich Street, 220084 Minsk, Belarus; baranovsky@iboch.by (A.V.B.); agilep@yahoo.com (A.A.G.); 2Institute of Biomedical Chemistry, Pogodinskaya Str. 10 Building 8, 119121 Moscow, Russia; leonid.kaluzhskiy@ibmc.msk.ru (L.A.K.); yuri.mezentsev@ibmc.msk.ru (Y.V.M.); 3G.B. Elyakov Pacific Institute of Bioorganic Chemistry, Far Eastern Branch, Russian Academy of Sciences, Pr. 100-let Vladivostoku 159, 690022 Vladivostok, Russia; makarieva@piboc.dvo.ru (T.N.M.); tabakmakher_km@piboc.dvo.ru (K.M.T.); ivanchina@piboc.dvo.ru (N.V.I.); paveldmt@piboc.dvo.ru (P.S.D.)

**Keywords:** lanosterol 14-alpha demethylase, CYP51, cytochrome P450, azole inhibitors, heterocyclic analogues of steroids, marine steroids, drug resistance, antifungal drugs, *Candida krusei*

## Abstract

Due to the increasing prevalence of fungal diseases caused by fungi of the genus *Candida* and the development of pathogen resistance to available drugs, the need to find new effective antifungal agents has increased. Azole antifungals, which are inhibitors of sterol-14α-demethylase or CYP51, have been widely used in the treatment of fungal infections over the past two decades. Of special interest is the study of *C. krusei* CYP51, since this fungus exhibit resistance not only to azoles, but also to other antifungal drugs and there is no available information about the ligand-binding properties of CYP51 of this pathogen. We expressed recombinant *C. krusei* CYP51 in *E. coli* cells and obtained a highly purified protein. Application of the method of spectrophotometric titration allowed us to study the interaction of *C. krusei* CYP51 with various ligands. In the present work, the interaction of *C. krusei* CYP51 with azole inhibitors, and natural and synthesized steroid derivatives was evaluated. The obtained data indicate that the resistance of *C. krusei* to azoles is not due to the structural features of CYP51 of this microorganism, but rather to another mechanism. Promising ligands that demonstrated sufficiently strong binding in the micromolar range to *C. krusei* CYP51 were identified, including compounds 99 (Kd = 1.02 ± 0.14 µM) and Ch-4 (Kd = 6.95 ± 0.80 µM). The revealed structural features of the interaction of ligands with the active site of *C. krusei* CYP51 can be taken into account in the further development of new selective modulators of the activity of this enzyme.

## 1. Introduction

Morbidity and mortality associated with fungal infections are constantly increasing [1,2]. This is largely due to an increase in the proportion of resistant fungal species, or the emergence of resistant strains in species that were initially sensitive to the action of antifungal drugs. The widespread use of antifungal agents, both in clinical settings and agriculture, has exerted selective pressure on fungal populations, facilitating the development of drug resistance. Additionally, factors such as immunocompromised patients, invasive medical procedures, and global travel contribute to the rising incidence of fungal infections. Candidiasis is an infectious disease caused by yeasts of the *Candida* species. Fungi of the genus *Candida* are responsible for about 80% of systemic mycoses in immunocompromised people [3,4]. Although *Candida albicans* (*C. albicans*) is the predominant cause of candidiasis, the proportions of other *Candida* species in infection have increased over the past two decades [5]. Other most important and common pathogens of the *Candida* genus include *Candida glabrata, Candida tropicalis, Candida parapsilosis,* and *Candida krusei* [6,7,8,9]. At the same time, the proportion of *Candida* species that are resistant to existing antifungals has recently increased significantly, and some strains are multidrug-resistant [2,8,10,11,12,13].

*Candida krusei* (*C. krusei*) is a yeast-like fungus of the genus *Candida* that causes opportunistic infections in humans. This yeast is generally regarded as a transient commensal fungus that inhabits mucous membrane of healthy individuals [4,14] *C. krusei* is almost completely resistant to azole antifungals (especially fluconazole and ketoconazole) and rapidly develops resistance to other antifungal drugs (e.g., anidulafungin, micafungin, 5-flucytosine, and amphotericin B) and is therefore is difficult to treat [2,5]. When resistance to not only azoles but also to other classes of antifungal drugs develops, a combination therapy is used. For example, the combined effect of amphotericin B and caspofungin [15], caspofungin and posaconazole [16], or isavuconazole and micafungin [17] have shown efficacy in overcoming the multi-drug resistance of the *Candida* species.

Azoles are broad-spectrum active pharmaceutical ingredients and are used as first-line drugs for the treatment of systemic mycoses or, as mentioned above, can also be part of combination therapy [18]. Azole antifungal drugs can be divided into two major classes (derivatives of imidazole and triazole derivatives) and three generations (Figure 1) [19].

The target of azoles is the key enzyme of ergosterol biosynthesis, steroid-14α-demethylase (CYP51), which belongs to the cytochrome P450 superfamily [20,21]. The emergence of resistance of pathogenic fungi to azoles and other existing antifungal agents has led to the need for new effective drugs with high specificity to resistant strains of fungi. Recently, special attention has been paid to compounds obtained from natural sources in order to reduce the toxicity and side effects of commercial drugs. Sherif T. S. Hassan and co-authors demonstrated the antifungal combinatory effect of plumbagin, a plant-derived compound, with amphotericin B against *C. albicans* clinical isolates [22]. C. F. Bezerra et al. revealed that flavonoid and tannic fractions of guava leaves have antifungal properties against *C. albicans*, *C. tropicalis,* and *C. krusei* [23]. Flavonoids from *Plinia cauliflora* leaves also showed inhibitory activity against *C. krusei* [24]. It should be noted that *C. krusei* CYP51 as a target enzyme of azoles is also in the purview of researchers. Fukuoka et al. investigated the interactions of fluconazole and voriconazole with the CYP51s of *C. krusei* and fluconazole-susceptible *C. albicans* and constructed homology models of these enzymes based on the crystal structure of CYP51 from *Mycobacterium tuberculosis* [25]. Thus, the study of the interaction of *C. krusei* CYP51 with potential inhibitors, particularly from natural sources, is of special interest. 

A correlation between the presence of mutations in the *ERG11* gene that encodes CYP51 and the development of resistance to azole drugs has been shown for *C. albicans* [26,27,28], *C. parapsilosis* [29], and *C. tropicalis* [30]. In the case of *C. krusei*, the role of CYP51 in the emergence of azole resistance remains unclear [25,31].

Therefore, the main aims of the study are to assess whether the azole resistance of *C. krusei* is associated with the structural features of *C. krusei* CYP51 or due to other factors and to identify new chemical scaffolds of *C. krusei* CYP51 active site ligands that can be used for the development of a new generation of inhibitors. 

The present work is devoted to the molecular cloning, purification, and evaluation of the ligand-binding properties of *C. krusei* CYP51.

## 2. Materials and Methods

### 2.1. Materials

The following reagents and reagent kits were used in this work: guanidine thiocyanate, potassium acetate, sodium acetate, Triton X-100, Tris(hydroxymethyl)aminomethane (Tris), and sodium ethylenediaminetetraacetate (Na-EDTA); P450 inhibitors, imidazole, and triazole derivatives (Sigma-Aldrich, St. Louis, MO, USA); agarose and dithiothreitol (GibcoBRL, Gaithersburg, MD, USA); dNTP mixture (Fermentas, Vilnius, Lithuania); thermostable RTaq DNA polymerase and amplification buffer (IBOCH NASB, Minsk, Belarus); restriction endonucleases (NEB, USA; Fermentas, Lithuania); benzonase (IBOCH NASB, Minsk, Belarus); Wizard™ Plus SV Minipreps DNA Purification System plasmid DNA isolation kit (Promega, Madison, WI, USA); NucleoSpin Gel and PCR Clean-up reagent kit for DNA extraction from agarose gel (Macherey-Nagel GmbH & Co. KG, Düren, Germany); In-Fusion HD EcoDry cloning kit (Clontech, Mountain View, CA, USA); BigDye™ Terminator v3.1 Cycle Sequencing Kit (Applied Biosystems, Foster City, CA, USA); BrilliantDye™ Terminator Cycle Sequencing Kit v. 3.1 (NimaGen, Nijmegen, The Netherlands); BigDye XTerminator Purification Kit DNA sequencing clean-up kit (Applied Biosystems, Foster City, CA, USA); yeast extract, peptone, tryptone (BD/Difco, Fisher Scientific, Franklin Lakes, NJ, USA), and arabinose (Sigma-Aldrich, St. Louis, MO, USA); acrylamide, ammonium persulfate, and bromophenol blue (Bio-Rad, Hercules, CA, USA), PMSF (Sigma-Aldrich, Taufkirchen, Germany); CHAPS (PanReac AppliChem, Darmstadt, Germany); IPTG, FeCl_3_, ZnCl_2_, CoCl_2_, Na_2_MoO_4_, CuCl_2_, H_3_BO_3_, NaOH, KCl, CaCl_2_, MgCl_2_, NaCl, MgSO_4,_ imidazole, and kanamycin (Glentham Life Sciences, Corsham, UK); β-mercaptoethanol (AppliChem, Darmstadt, Germany); KH_2_PO_4_ (Fisher Bioreagents, Brussels, Belgium); K_2_HPO_4_, SDS (Sigma-Aldrich, Taufkirchen, Germany); Coomassie R-250, sodium isocitrate, isocitrate dehydrogenase, α-aminolevulinic acid, NADPH, and glycerin (Sigma-Aldrich, Taufkirchen, Germany); DEAE-sepharose (Pharmacia & Upjohn, North Peapack, NJ, USA); hydroxyapatite Bio-Gel HTP (Bio-Rad, Hercules, CA, USA); Protino Ni-NTA-agarose (Macherey-Nagel GmbH & Co. KG, Düren, Germany); sinapic acid (Bruker Daltonics, Bremen, Germany); formic acid (Sigma-Aldrich, St. Louis, MO, USA); trifluoroacetic acid (Acros Organics, Morris Plains, NJ, USA); and glacial acetic acid (Fisher Chemical, Loughborough, UK). 

### 2.2. Modeling of C. krusei CYP51 3D Structure

The model structure of *C. krusei* CYP51 was predicted using AlphaFold2 in ColabFold v1.5.2 [32]. As a result, five possible structural hypotheses were generated and the top ranked structure was relaxed using Amber in ColabFold with the default settings. The MSA options were mmseqs2_unirif_env for msa_mode and unpaired_paired for pair_mode.

The model structure of *C. krusei* CYP51 was compared with the structure of *C. albicans* CYP51 (5V5Z) because their sequences have high similarity (75.3%). For this purpose, we used the tool MatchMaker of the software UCSF Chimera v1.15 [33] with Needleman-Wunsch as the alignment algorithm and BLOSUM-62 as the matrix.

### 2.3. Molecular Cloning and Expression of C. krusei CYP51

DNA for cloning was isolated from a clinical sample exhibiting resistance to antifungal drugs using a kit of reagents for DNA extraction—“DNA-VK”, developed at the Institute of Bioorganic Chemistry of the National Academy of Sciences of Belarus. The clinical sample was isolated at the Belarussian Research Center for Pediatric Oncology, Hematology and Immunology. The pCW-lic plasmid vector (Structural Genomics Consortium, Toronto) and the GroES-GroEL complex of molecular chaperones were used for the cloning of the PCR products and subsequent expression in bacterial cells. Strains used for cloning and expression were *E. coli* DH5α and *E. coli* C41 (DE3), respectively (Invitrogen, Carlsbad, CA, USA).

For the molecular cloning, the following primers containing two restriction sites, NdeI and HindIII, and 6 His residues were designed and used:*C. krusei*_fusF: ttaggaggtcatatgcatcatcatcatcatcattccgtcatcaaggcaat;*C. krusei*_fusR: taagcttcgtcatcagttcttttgtcttctctc.

PCR products of *C. krusei* CYP51 (CkrCYP51) cDNA were ligated into the pCW-lic expression vector using the In-Fusion HD EcoDry cloning kit (Clontech, Mountain View, CA, USA). A new expression construct (pCW-lic_CkrCYP51) was sequenced to confirm the sequence.

*E. coli* C41 (DE3) clones carrying the expression vector (pCW-lic_CkrCYP51) and the GroES-GroEL molecular chaperone complex were used to inoculate 5 mL of LB medium containing ampicillin (100 µg/mL) and kanamycin (35 µg/mL). The cell culture was incubated at 37 °C with constant shaking at 180 rpm overnight. Flasks with 100 mL of TB medium containing potassium phosphate buffer (pH 7.5), trace elements, ampicillin (100 μg/mL), and kanamycin (35 μg/mL) were inoculated with 2.5 mL of the overnight culture. The cell culture was incubated at 37 °C with constant shaking at 180 rpm until the culture absorbance at 600 nm (A_600_) was 2.5. Subsequently, 200 mL of cell culture was inoculated into flasks with 800 mL of fresh TB medium (total volume was 1 L) containing potassium phosphate buffer (pH 7.5), trace elements, ampicillin (100 μg/mL), and kanamycin (35 μg/mL). The cell culture was transferred into large flasks and incubated at 37 °C with constant shaking at 180 rpm until the culture absorbance at 600 nm was 0.6–0.8. CYP51 expression was induced by adding IPTG (0.5 mM) and aminolevulinic acid (0.65 mM) to the culture. Bacterial cells were incubated at 37 °C with constant shaking at 120 rpm for 48 h. The cells were collected by centrifugation at 3000× *g* for 20 min at 4 °C. The cell pellet was resuspended in 50 mM potassium phosphate buffer (pH 7.4) containing 20% glycerol, 0.1 mM PMSF, 0.3 M NaCl in a ratio of 1:4 (4 mL of buffer per 1 g of cells), and frozen at −78 °C.

### 2.4. Purification of C. krusei CYP51

Cells pellets were thawed and 0.1 mM PMSF and CHAPS ionic detergent were added to a final concentration of 0.2%, followed by cell homogenization. CYP51 was solubilized from the membranes by adding CHAPS ionic detergent to the bacterial cell suspension to a final concentration of 1%. The detergent was added slowly, drop by drop, with constant stirring at 4 °C. The suspension was centrifuged at 4 °C for 1 h at 25,000 rpm to pellet the membranes. The supernatant was applied to a Ni NTA-agarose column equilibrated in buffer A containing 50 mM potassium phosphate buffer (pH 7.4), 20% glycerol, 0.3 M NaCl, and 0.5 mM PMSF. The column was then washed with 20 bed volumes of buffer B (25 mM imidazole, 20% glycerol, 0.3 M NaCl, 50 mM potassium phosphate buffer (pH 7.4), 0.2% CHAPS, 1.4 mM β-mercaptoethanol). The protein was eluted with the addition of 250 mM imidazole to buffer B. Colored fractions were collected followed by 10 times dilution with 5 mM potassium phosphate buffer (pH 7.4) containing 0.3 M NaCl, 20% glycerol, 0.2% CHAPS, and 0.1 mM dithiothreitol (buffer C) and then applied to a hydroxyapatite (HAP) column. The column was pre-washed with 5 bed volumes of buffer C. After application of the diluted eluate, the column was washed with 20 bed volumes of buffer C containing 10 mM potassium phosphate buffer (pH 7.4). The protein was eluted from the column by increasing the concentration of potassium phosphate to 500 mM. The collected fractions of recombinant protein were stored at −78 °C.

The concentration of CYP51 was determined from the difference spectrum of the reduced CO complex minus the reduced hemeprotein using the extinction coefficient e450–490 = 91 mM^−1^ cm^−1^ [34,35].

### 2.5. Reconstitution Activity Assay

The reconstitution activity assay was conducted in accordance with the methodology outlined by Kaluzhskiy in 2021 [36]. To determine the activity of CkrCYP51, the following experimental conditions were used: a temperature of 37 °C and a buffer solution of 50 mM potassium phosphate (pH 7.4), supplemented with 4 mM magnesium chloride (MgCl_2_) and 0.1 mM dithiothreitol (DTT), in the presence of lipids. The lipid mixture comprised L-α-dilauroyl-sn-glycero-3-phosphocholine, L-α-dioleoyl-sn-glycero-3-phosphocholine, and L-α-phosphatidyl-L-serine, each at a concentration of 0.15 mg/mL in a 1:1:1 ratio.

The final concentrations of the enzymes, CkrCYP51 and CPR, were 1.0 µM and 4.0 µM, respectively. The recombinant proteins and lipids were mixed and preincubated for 5 min at room temperature. Subsequently, lanosterol, as a stock solution of 10 mM in ethanol, was added to the reaction mixture to achieve a final concentration of 50 µM. After a further 10 min preincubation at 37 °C, the reaction was initiated by introducing NADPH to a final concentration of 0.25 mM. At specific time intervals, 0.5 mL samples were withdrawn from the incubation mixture.

To extract steroids, 5 mL of ethyl acetate was employed, and the mixture was vigorously agitated. The resulting mixture was separated into water and organic phases through centrifugation at 3000 rpm for 10 min. The organic layer was then carefully collected and dried under an argon flow. Subsequently, 50 µL of methanol was added to the pellet, and the steroids were analyzed using an Agilent 1200 series HPLC instrument (Agilent Technologies, Santa Clara, CA, USA) equipped with an Agilent Triple Quad 6410 mass spectrometer (Agilent Technologies).

The analysis was performed using a Zorbax Eclipse XDB C18 column (4.6 × 150 mm; 5 µm) (Agilent Technologies) using a gradient elution method. Mobile phase A consisted of 0.1% (*v/v*) formic acid (FA) in water, while mobile phase B comprised 0.1% (*v/v*) FA in a methanol/1-propanol mixture (75:25, *v/v*). The gradient involved a transition from 75% to 100% B over 5 min, with a flow rate of 500 µL per minute. The column temperature was maintained at 40 ± 1 °C. Mass spectrometry experiments were conducted with an atmospheric pressure chemical ionization source (APCI) in positive ion mode. The APCI settings included a gas temperature of 200 °C, vaporizer temperature of 350 °C, gas flow rate of 7 L/min, nebulizer pressure of 40 psig, Vcap at 4000 V, corona at 4 µA, and fragmentor at 100 V. Data acquisition was performed in the MS2Scan mode, covering a mass range from 200 to 550 Da.

### 2.6. Ligand Screening

A ligand screening procedure was used to identify possible ligands for CkrCYP51 among natural and synthesized steroids.

In this experiment, 1 μL of the test compound solution was diluted in 100% DMSO to within a concentration of 0.01 M (the final concentration of the compound in solution was 40 μM) was added to 250 μL of the protein solution (the final protein concentration in solution was 3 μM). Then, the absorption spectrum was recorded in the range of 350–500 nm using a Clariostar Plus microplate reader (BMG Labtech, Ortenberg, Germany). A 40 μM solution (the final concentration in the well) of the model substrate and the model type II inhibitor for the analyzed cytochrome P450 were added to the last wells of the plate as controls. For ligands that demonstrated binding to the enzyme, the dissociation constants of the complexes containing CYP51 and steroid derivatives were identified using the spectrophotometric titration method.

### 2.7. Spectrophotometric Titration

Dissociation constants of complexes of CYP51 with the substrate, inhibitors, and potential ligands were determined by the spectrophotometric titration method in [37]. Briefly, a protein solution (final concentration of 1 μM) in buffer (50 mM potassium phosphate buffer, pH 7.4) was placed in two cuvettes (1 × 1 cm), and a baseline was recorded on a spectrophotometer at a wavelength of 350–500 nm. The spectra were detected with a Cary 5000 UV-VIS-NIR spectrophotometer. A ligand solution was added to the experimental cuvette and an equal volume of solvent was added to the control cuvette; the change in optical absorption was recorded in the same wavelength range. The solutions in the cuvettes were thoroughly mixed by pipetting after adding the ligand solution or solvent and incubated for 1–2 min. The maximum and minimum absorption values were registered.

The dissociation constant of the enzyme–ligand complex (Kd) was used as a parameter characterizing the strength of ligand binding in the active site. Kd was calculated by approximating the spectrophotometric titration data (the absorbance changes in the difference spectra versus the concentration of the free ligand) with a nonlinear function (1) [38] or the Hill Equation (2) [36], implemented in the Origin 2016 program. Levenberg–Marquardt algorithm was used for the fitting in both cases:(1)$$A={\Delta A}_{max}\frac{\left[S\right]+{\left[E\right]}_{0}+K_{d}-\sqrt{{\left([S]+{\left[E\right]}_{0}+K_{d}\right)}^{2}-4{\left[E\right]}_{0}[S]}}{2{\left[E\right]}_{0}}$$
(2)$$A={\Delta A}_{max}\times \left(\frac{S\times n}{K_{d}\times n+S\times n}\right)$$
where: 

A—the observed absorption;

ΔA_max_—the absorbance change at ligand saturation;

E_0_—the protein concentration;

S—the ligand concentration;

K_d_—the dissociation constant for the enzyme–ligand complex;

n—a Hill coefficient.

### 2.8. Mass Spectrometric Analysis

Mass spectrometric analysis was carried out according to Bruker Daltonik protocols for a Microflexs LRF series time-of-flight mass spectrometer. A polished steel target containing 96 wells was employed for mass spectrometry analysis (MSP Polished Target Steel, 96, Bruker Daltonik, Bremen, Germany). Sample and matrix solutions were applied to the target by the dried droplet method. A 1 µL volume of fresh saturated matrix solution (sinapinic acid, SA) and 1 µL of sample solution were applied to a clean, free well and were resuspended directly in the target. After complete drying of the mixture of matrix and sample (5–10 min), the mass spectrometric analysis was carried out.

Matrix-assisted laser desorption ionization mass spectrometry (MALDI-MS) of the sample was performed on a time-of-flight mass spectrometer of the Microflexs LRF series (Bruker Daltonics, Bremen, Germany), in the linear mode of detecting positive ions in the *m/z* range from 10,000 Da to 90,000 Da. Protein Calibration Standard II (Bruker Daltonics, Bremen, Germany) was used as a standard.

### 2.9. Obtaining of Chlorotopsenthiasterol Sulfate D (S-232-Cl) and Granulatoside A (Ch-4)

Chlorotopsenthiasterol sulfate D was isolated from the marine sponge Halichondria vansoesti [39]. Granulatoside A was isolated from the starfish Choriaster granulatus [40].

## 3. Results

### 3.1. Comparative Analysis of CYP51 of Clinically Relevant Fungi of the Genus Candida

In order to establish the nature of *C. krusei*’s resistance to azole-class antifungals, the primary structure of CYP51 of pathogenic fungi of the genus *Candida* was analyzed (Figure 2).

According to the amino acid sequence alignment of CYP51 proteins, *C. krusei* CYP51 contains only two amino acids corresponding to the amino acid substitutions found in CYP51 azole-resistant strains of *C. albicans*—L261 (corresponding to the M258L mutation of *C. albicans* CYP51) and I409 (corresponding to the V404I mutation of *C. albicans* CYP51). In this case, L261 is located in the α-helix G and in close proximity to the FG-loop, which is involved in the process of binding substrates and inhibitors and belongs to SRS-3 (Substrate Recognition Region-3) (Figure 2). I409 is located in close proximity to the “meander region” responsible for the interaction with the redox partner of CYP51. It should be noted that for the M258L and V404I mutations of CYP51 *C. albicans,* it has not been shown that they directly lead to the development of resistance. However, these mutations only occur, along with others, in resistant strains [26,27].

Also, a 3D comparative analysis of the crystal structure of *C. albicans* CYP51 and the spatial structure model of *C. krusei* CYP51 was carried out (Figure 3).

The structure of *C. krusei* CYP51 was predicted using AlphaFold2 and relaxed using Amber in ColabFold v1.5.2. The acceptable quality of the model was confirmed by the high scores of pLDDT (90.2) and pTM (0.885). 

The model of *C. krusei* CYP51 and the structure of *C. albicans* CYP51 5V5Z were 3D aligned in UCSF Chimera. The RMSD between 452 pruned atom pairs was shown to be 0.758 angstroms, indicating a high similarity between the three-dimensional structures of the proteins and superpositions of their secondary structures.

A comparative analysis of the 3D structures of CYP51 of *C. krusei* and *C. albicans* (Pdb Id: 5V5Z) showed that these enzymes differ slightly in the region that is bound by azole-type inhibitors (Figure 3B,C). Of all the amino acid residues that form the entrance to the active site, only two pairs can be distinguished that differ between the two structures: S506 (*C. albicans*)–T507 (*C. krusei*) and I231 (*C. albicans*)–L234 (*C. krusei*). In *C. albicans* CYP51, S506 is involved in the formation of a hydrogen bond with an oxygen (O7) of the itraconazole in the active center. Homology models of CYP51 from *C. krusei* and *C. albicans* were previously constructed using the crystal structure of CYP51 from *Mycobacterium tuberculosis*, with fluconazole and voriconazole as ligands in the active site [25]. The authors identified several amino acids located within 12 Å of the ligand bound in the active site that are different between these two enzymes. However, according to the authors’ model, none of these amino acids were involved in the interaction with the azole in the active site. Our results align with this observation, as none of the amino acids examined by the authors were implicated in the interaction with itraconazole.

Thus, a comparative analysis of the structures of *C. krusei* CYP51 and CYP51 of other clinically relevant species of *Candida* does not allow us to unequivocally state that the resistance of *C. krusei* to azoles is due to the peculiarities of the CYP51 structure of this microorganism. To establish the properties of this enzyme, molecular cloning and analysis of the ligand-binding properties of *C. krusei* CYP51 were carried out.

### 3.2. Molecular Cloning, Expression, and Purification of C. krusei CYP51

In order to obtain recombinant CkrCYP51, the molecular cloning of CkrCYP51 cDNA into the expression vector pCW-lic was carried out (Figure 4). *C. krusei* genomic DNA isolated from a clinical specimen showing resistance to antifungal drugs was used as a cDNA source. The sequencing results of the novel expression construct revealed that the *ERG11* gene (that encodes for CYP51) of this clinical isolate contains the K136R mutation that is not present in either the reference sequence of C.krCYP51 (XP_029322955) or in the sequence of the clinical isolate studied in the previous work [25]. It is important to note that the location of this substitution corresponds to the region of “hot spot” mutations for the development of resistance in *C. albicans* (Figure 2). Moreover, this region corresponds to SRS-1 of CYP51. This suggests that this substitution may influence the development of resistance to antifungal drugs.

Further, we carried out heterologous expression and purification of the recombinant protein CkrCYP51 from a bacterial cell mass. To simplify the procedure for isolating the recombinant hemoprotein by metal affinity chromatography, six additional histidine residues were introduced into the N-terminal sequence of CkrCYP51. The highly purified recombinant enzyme has the characteristic spectral properties of the six-coordinated low-spin form of cytochrome P450: an absorption maximum of 280 nm from the protein part, an absorption maximum at 417 nm from the prosthetic group in the Soret band region, and two characteristic maxima in the region of the α and β bands at 535 and 567 nm (Figure 5a).

Spectrophotometric analysis of the reduced form of C.krCYP51 in a complex with carbon monoxide demonstrated the formation of a characteristic carbonyl complex with an absorption maximum at 447 nm (Figure 5b). The results of gel electrophoresis under denaturing conditions confirm the homogeneity and high purity of the recombinant C.krCYP51. The isolated protein by weight corresponds to the theoretically calculated ~61.39 kDa (Figure 5c).

MALDI-TOF-mass-spectrometric analysis showed that recombinant C.krCYP51 has a molecular weight of 61,190.63 Da. The data obtained are consistent with the value calculated theoretically based on the known amino acid composition and the introduced modifications (tag of six histidine amino acid residues) (Figure 6).

In an in vitro reconstituted system, C.krCYP51 exhibited 14-α-demethylase catalytic activity to produce the demethylated lanosterol product 4,4-dimethylcholesto-8,14,24-trienol (FF-MAS).

As a result of molecular cloning, expression, and purification, we obtained CYP51 *C. krusei* for the first time.

### 3.3. Ligand-Binding Properties of CYP51 C. krusei

Considering the trend towards increasing incidences of fungal diseases (both superficial and severe visceral mycoses associated with HIV infection and oncohematological diseases), the development of pathogen resistance to existing drugs, and the identification of fungal species that were previously considered non-pathogenic (currently, about 400 fungal species are considered potential pathogens), the need to search for new effective antifungal agents has increased. Since there are various mechanisms for the development of resistance to azole inhibitors for cytochromes P450, including those not associated with the ligand-binding features of CYP51 of fungi, we first evaluated the binding of known antifungal drugs and compounds with fungicidal action to the active site of C.krCYP51.

To assess the ligand-binding properties of C.krCYP51, we performed spectrophotometric titration. This method is based on the fact that when interacting with ligands, cytochromes P450 demonstrate a characteristic spectral response, reflecting a change in the spin state of the heme iron. Depending on the type of ligand, three main types of spectral changes can be detected, which are characterized by a maximum and a minimum in the differential absorption spectrum at certain wavelengths (type I, type II, and reverse type I spectral response; the latter is sometimes called modified type II) [53].

#### 3.3.1. Interaction of *C. krusei* CYP51 with Azole Antifungals

Over the past two decades, drugs from the azole group, which are inhibitors of sterol-14α-demethylase, have been widely used in the treatment of fungal infections [54,55].

In order to assess the interaction of C.krCYP51 with the main inhibitors, spectrophotometric titration of the recombinant protein with various azoles was performed. The spectrophotometric titration data allowed us to determine the interaction parameters of the enzyme with inhibitors and to calculate the dissociation constant of the CYP51–azole complexes. (Table 1). A type II spectral response was detected when azoles interacted with CYP51.

Table 1 shows that the most effective binding of this enzyme form is observed for clotrimazole (Kd = 0.0013 ± 0.0008 µM) and econazole (Kd = 0.0013 ± 0.0004 µM), which are broad-spectrum antimycotics and are used to treat local infections. The rest of the azoles also have a rather high affinity for C.krCYP51.

That is, despite the fact that *C. krusei* has multiple resistance to antifungal drugs, the isolated recombinant C.krCYP51 shows a fairly high affinity for drugs that are currently widely used in antifungal therapy. Based on the experimental data obtained in this work, it can be concluded that the development of resistance to azoles may be due to other mechanisms not related to CYP51. In the case of *C. krusei*, these mechanisms may include increased expression of the *ERG11* gene; drug excretion due to overexpression of transporters encoded by the *ABC1*, *ABC2*, *ABC11*, and *ABC12* genes; or mutations in the *ERG3* gene (that encodes the enzyme steroid-Δ5,6-desaturase, which is also involved in the biosynthesis of ergosterol) [2].

In this regard, the search for potential inhibitors of C.krCYP51 that have a chemical scaffold different from that of known azole and triazole inhibitors of CYP51 is of particular relevance.

#### 3.3.2. Interaction of *C. krusei* CYP51 with Steroid Derivatives

The emergence of resistance of pathogenic fungi to existing antifungal drugs has led to the need for new effective drug compounds with a high specificity to resistant strains of fungi. This pressing issue not only underscores the importance of ongoing research and development in the field of antifungal medications but also highlights the significance of identifying novel drug targets and therapeutic approaches. In this context, interdisciplinary efforts involving molecular biology, genomics, and medicinal chemistry play a crucial role in the discovery of innovative antifungal agents that can combat drug-resistant strains, ultimately ensuring more effective treatments and improved outcomes for patients facing fungal infections.

Compounds that are structurally similar to the natural substrate of CYP51—lanosterol—were synthesized by the Institute of Bioorganic Chemistry of the National Academy of Sciences of Belarus. Aside from that, derivatives of steroids from marine organisms that have different physiological activities (antiviral, antimicrobial, analgesic, anti-inflammatory, hypotensive, and others [56]) were isolated by the G. B. Elyakov Pacific Institute of Bioorganic Chemistry, Far Eastern Branch of the Russian Academy of Sciences. 

These compounds have a unique structure and were tested as potential ligands for C.krCYP51 since the functional features of this cytochrome indicate the chemical complementarity of these natural and synthesized compounds to the active center of the enzyme.

The primary objective of the experiment was to identify steroids with the strongest affinity for the active site of C.krCYP51. These specific compounds had not been previously considered as potential ligands for this enzyme. It was crucial to establish a range of compounds suitable for potential modifications and to investigate the types of alterations that could lead to the discovery of the most effective inhibitors.

Initially, screening studies were carried out. Some compounds for screening were identified using the SPR (surface plasmon resonance) method in our previous work [57]. A total of 80 steroid derivatives were tested. The analysis revealed the binding of some compounds to C.krCYP51. 

In the next step, spectrophotometric titration was carried out to determine the affinity (Kd) for the enzyme–ligand complexes of the test compounds with C.krCYP51. Binding to the active site was demonstrated by four compounds (Table 2). When the studied ligands interacted with C.krCYP51, type I spectral changes were detected.

The best binding was demonstrated by compound 99 (3β,20-dihydroxy-24-hydroxyiminocholest-5,22-diene) with a Kd of 1.02 ± 0.14 µM, which can be used for further development of selective inhibitors of CYP51. A lower affinity was noted for compounds 73c (14,17-etheno-3-hydroxy-16α-nitroestra-1,3,5(10)-trien-17β-yl acetate) and Ch-4 (granulatoside A) (Kd = 10.7 ± 1.5 µM and Kd = 6.95 ± 0.80 µM, respectively). The compound S-232-Cl (chlorotopsenthiasterol sulfate D) showed the lowest binding to the active site of C.krCYP51.

## 4. Discussion

CYP51 enzymes found in pathogenic fungi serve as essential drug targets for the development of selective antifungal agents aimed at addressing azole-resistant fungal infections. The existing azole antifungal drugs have a wide spectrum of action on enzymes of various families of human cytochromes P450, causing diverse side effects [58]. Therefore, the development of alternative fungal cytochrome P450 inhibitors remains relevant. 

In this regard, in order to establish the molecular mechanisms of resistance, as well as to carry out the initial stages of the rational drug design of a new generation of antifungal drugs, it is necessary to:-Obtain highly purified proteins of pathogenic organisms, especially resistant strains, that are targets for the action of drug compounds;-Investigate the interaction of molecular drug targets with known drugs;-Search for potential ligands of molecular targets that have a chemical scaffold different from that of existing drugs.

One of the mechanisms of azole resistance in the *Candida* genus is the presence of a mutation in the *ERG11* gene that encodes CYP51 [58]. Previous studies have shown a correlation between mutations in CYP51 and the development of resistance to azole antifungals [26,27,28] for species such as *C. albicans*, *C. parapsilosis* [29], and *C. tropicalis* [30]; however, for *C. krusei*, the role of CYP51 in the acquisition of azole resistance remains uncertain [25,31].

In this work, we performed a comparative analysis of the structures of the CYP51s of *C. krusei* and *C. albicans.* Our findings revealed that the structures of these enzymes have no significant differences in the region that is bound by azole-type inhibitors. Only two amino acid residues the sequence that forms the entrance to the active site differ between the two structures: S506 (*C. albicans*)–T507 (*C. krusei*) and I231 (*C. albicans*)–L234 (*C. krusei*). In *C. albicans* CYP51, S506 is involved in the formation of a hydrogen bond with an oxygen (O7) of the itraconazole in the active center. The results of the present research are consistent with the data obtained in a previous work by Fukuoka et al. [25]. The structural models constructed earlier and obtained in this study correlate with each other. Those amino acids that differ between the CYP51s of *C. krusei* and *C. albicans* but do not participate in the interaction with the azoles (fluconazole and voriconazole) in the active site in the model structures obtained by Fukuoka et al. are also not involved in the interaction with itraconazole according to our models.

Thus, a comparative analysis of the structures of *C. krusei* CYP51 and *C. albicans* CYP51 does not allow us to definitely state that the resistance of *C. krusei* to azoles is due to the unique features of the CYP51 structure of this microorganism. 

An analysis of the ligand-binding properties of the isolated recombinant *C. krusei* CYP51 with azole inhibitors demonstrated a fairly high affinity for current antifungal drugs despite the fact that *C. krusei* exhibits intrinsic resistance to fluconazole and can acquire resistance to other azoles. Based on our experimental data, we can suggest that the resistance to azoles in *C. krusei* is driven by mechanisms other than CYP51 point mutations. Previous studies have considered mechanisms such as increased expression of the *ERG11* gene; drug excretion due to overexpression of transporters encoded by the *ABC1*, *ABC2*, *ABC11*, and *ABC12* genes; and mutations in the *ERG3* gene (that encodes the enzyme steroid-Δ5,6-desaturase, which is also involved in the biosynthesis of ergosterol) [2,59,60,61].

In addition, we assessed the interaction between *C. krusei* CYP51 and steroid derivatives. We have identified promising ligands that exhibit strong binding in the micromolar range to *C. krusei* CYP51 (designated as 99 and Ch-4).

The strong affinity of these ligands for the enzyme can be attributed to their structural resemblance to the natural substrate of the target protein, lanosterol, which shares a steroid core as its foundation. The binding of these ligands occurs through multiple van der Waals interactions with the hydrophobic side chains of amino acid residues.

For fungal CYP51, a relationship has been observed between the length of the ligand molecule and its ability to inhibit the enzyme [62]. Longer structures have the capability to establish additional contacts around the substrate access channel, thereby stabilizing its closed state. This clarifies why *C. krusei* CYP51 shows the highest affinity for the compound with the longer substituent at position 17—known as 99.

These identified patterns are crucial for gaining a deeper understanding of how ligands interact with the active site of *C. krusei* CYP51 and can be utilized in the development of selective modulators of its activity. Moreover, all the aforementioned compounds can serve as a starting point for the subsequent exploration of selective fungal CYP51 inhibitors.

## 5. Conclusions

A comparative analysis of the structures of *C. krusei CYP51* and the CYP51s of other clinically significant *Candida* species, as well as the ligand-binding properties of *C. krusei* CYP51 with azole-containing antifungal agents, indicates that the resistance of *C. krusei* to azoles is not due to the structural features of the CYP51 of this microorganism, but rather due to another mechanism. In an effort to find new scaffolds for CYP51 inhibitors, we have identified a group of natural and synthetic steroidal ligands of the *C. krusei* CYP51 active site that can be further explored to develop a new type of antifungal drug targeted at CYP51.

## Figures and Tables

**Figure 1 biomedicines-11-02873-f001:**
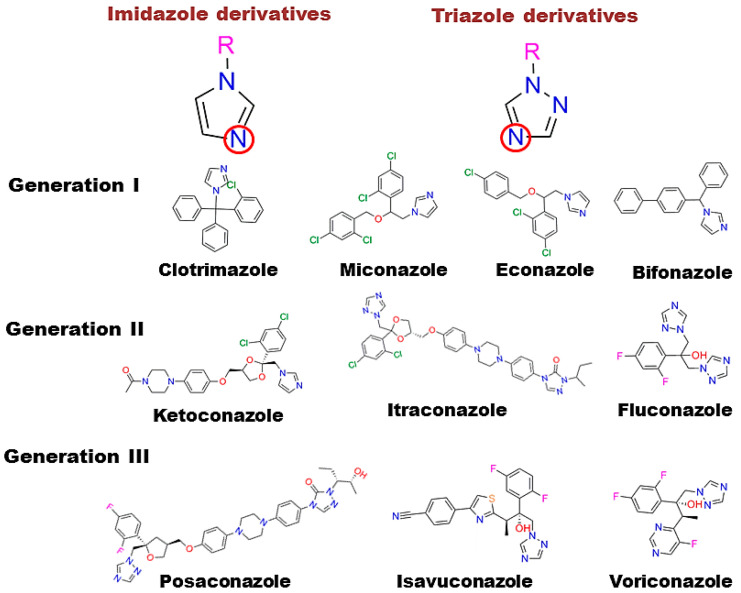
Classification of azole drugs. The nitrogen atom that forms a bond with the heme iron is marked with a red circle.

**Figure 2 biomedicines-11-02873-f002:**
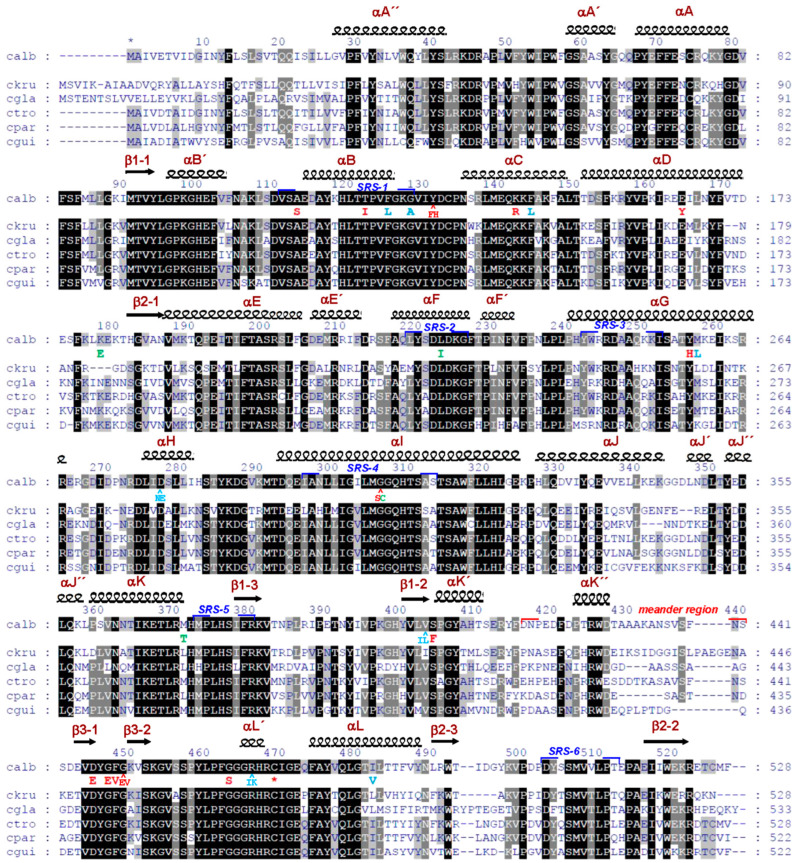
The multiple sequence alignment of CYP51 proteins from fungi of the genus *Candida*. Alignment was performed using the GeneDoc v.2.6 program. The assignment of secondary structure elements is based on the *Candida albicans* CYP51 structure (PDB: 5V5Z). White characters on the black background show 100% identity, white characters on the grey background show 75% similarity, and black characters on the grey background show 50% similarity. Mutations in azole-resistant *C. albicans* clinical isolates are highlighted and mutated residues are shown below the sequence (red colored—mutations attributed to resistance [27,28,41,42,43,44,45]; blue colored—resistance exhibited in combination with other mutations [27,28,44,46,47,48,49,50]; green colored—mutations we detected for the first time in azole-resistant *C. albicans* clinical isolate [51,52]). The numbers on top are shown for CYP51 *Candida albicans*. The red star shows a heme-bound cysteine. Blue frames indicate Gotoh’s SRSs. The amino acid sequences were taken from the NCBI: calb (*Candida albicans*—XP_716761); cgla (*Candida glabrata*—XP_445876); ctro (*Candida tropicalis*—XP_002550985); ckru (*Candida krusei/Pichia kudriavzevii*/—XP_029322955); cpara (*Candida parapsilosis*—ACT67904); cgui (*Candida guilliermondii/Meyerozyma guilliermondii*/—XP_001484034).

**Figure 3 biomedicines-11-02873-f003:**
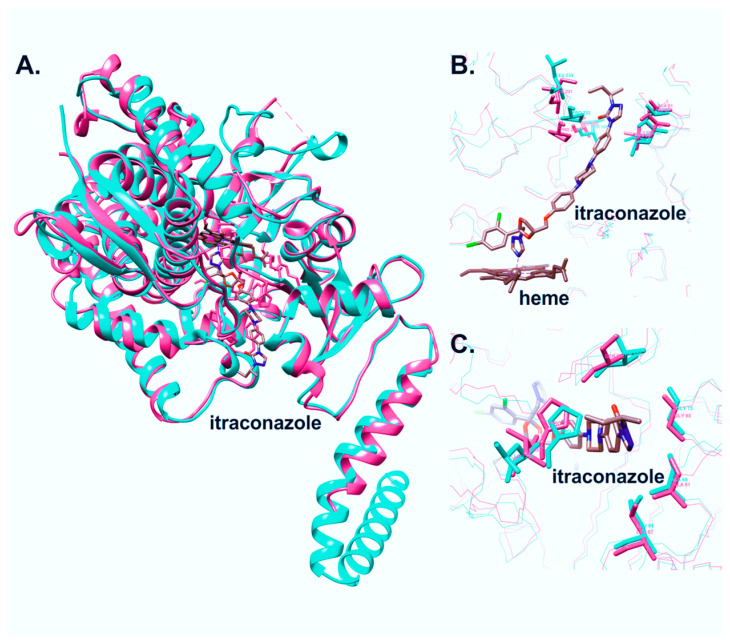
Three-dimensional alignment of CYP51 structures from *Candida albicans* (magenta) and *Candida krusei* (cyan). Overall folding (**A**), active site (**B**), and access to the substrate channel (**C**).

**Figure 4 biomedicines-11-02873-f004:**
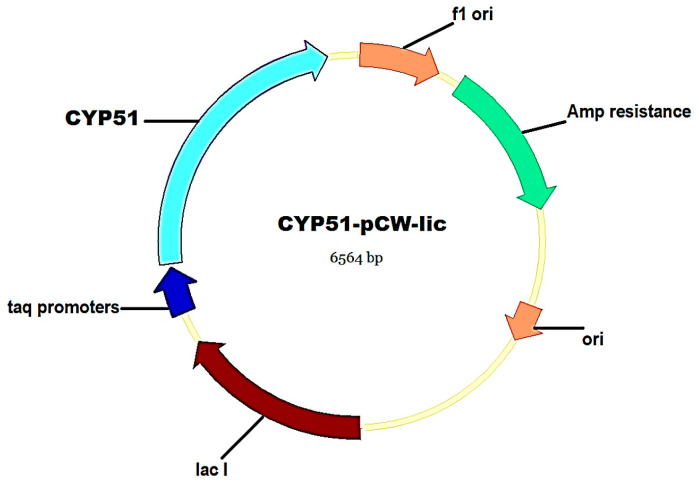
Map of CYP51-pCW-lic expression vector.

**Figure 5 biomedicines-11-02873-f005:**
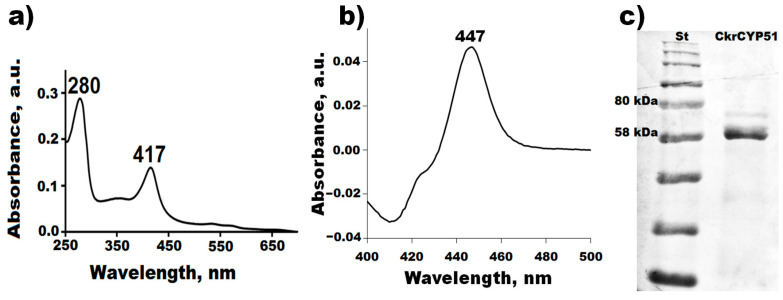
Absorption spectrum of the purified C.krCYP51 (**a**); difference spectrum of the carbonyl complex of reduced C.krCYP51 (**b**); results of electrophoretic analysis of purified C.krCYP51 in 12% PAGE under denaturing conditions. St (P7712, NEB)—molecular weight standard; 1—C.krCYP51 fraction after purification on HAP (**c**).

**Figure 6 biomedicines-11-02873-f006:**
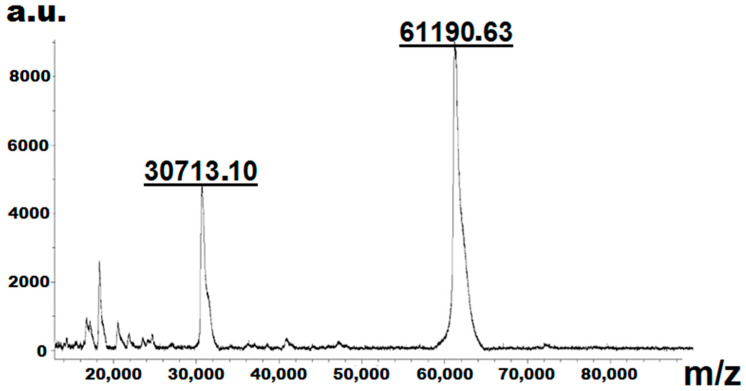
Results of mass spectrometric analysis of purified CkrCYP51.

**Table 1 biomedicines-11-02873-t001:** Interaction parameters of C.krCYP51 with azoles.

Compound	Structure	Spectrum	Kd (µM), ΔA_max_, Type of Spectral Response
Ketoconazole	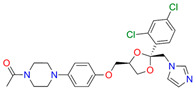	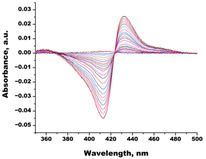	Kd = 0.0054 ± 0.0026 Titration range: 0.02–2.26 µM ΔA_max_ = 0.070 type II
Bifonazole	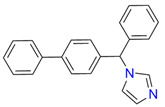	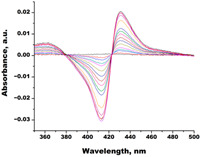	Kd = 0.003 ± 0.001 Titration range: 0.02–1.22 µM ΔA_max_ = 0.050 type II
Clotrimazole	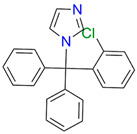	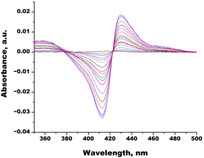	Kd = 0.0013 ± 0.0008 Titration range: 0.02–1.14 µM ΔAmax = 0.051 type II
Econazole	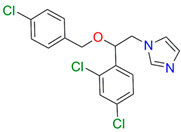	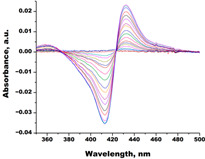	Kd = 0.0013 ± 0.0004 Titration range: 0.004–0.83 µM ΔA_max_ = 0.058 type II
Fluconazole	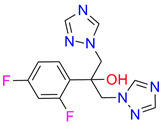	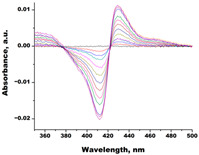	Kd = 0.004 ± 0.001 Titration range: 0.02–1.08 µM ΔA_max_ = 0.031 type II
Miconazole	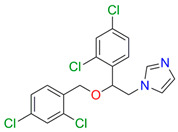	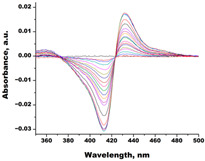	Kd = 0.002 ± 0.0009 Titration range: 0.02–1.48 µM ΔA_max_ = 0.048 type II
Voriconazole	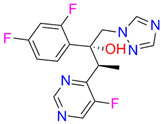	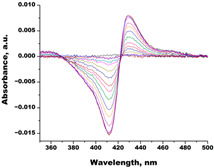	Kd = 0.01 ± 0.006 Titration range: 0.02–3.12 µM ΔA_max_ = 0.023 type II

Lines of different colors indicate spectra at different ligand concentrations.

**Table 2 biomedicines-11-02873-t002:** Interaction parameters of C.krCYP51 with steroid derivatives.

Compound	Structure	Spectrum	Kd, µM; ΔA_max_; Type of Spectral Response
Lanosterol	** 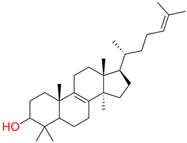 **	** 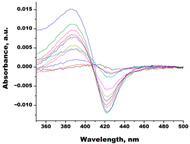 **	Kd = 2.82 ± 0.18 Titration range: 0.1–30.9 µM ΔA_max_ = 0.031 type I
73c (14,17-etheno-3-hydroxy-16α-nitroestra-1,3,5(10)-trien-17β-yl acetate)	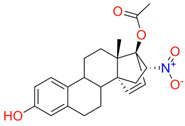	** 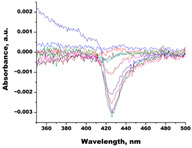 **	Kd = 10.7 ± 1.5 Titration range: 0.1–44.7 µM ΔA_max_ = 0.0069 type I
99 (3β,20-dihydroxy-24-hydroxyiminocholest-5,22-diene)	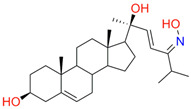	** 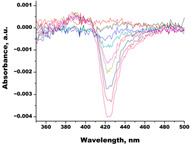 **	Kd = 1.02 ± 0.14 Titration range: 0.1–10.7 µM ΔA_max_ = 0.0047 type I
S-232-Cl (chlorotopsenthiasterol sulfate D, sponge *Halichondria vansoesti*)	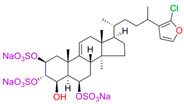	** 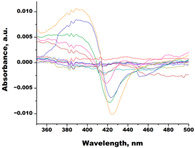 **	Kd = 21.23 ± 2.83 Titration range: 0.1–140.5 µM ΔA_max_ = 0.0034 type I
Ch-4 (granulatoside A, starfish *Choriaster* *granulatus*)	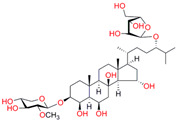	** 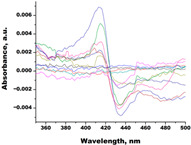 **	Kd = 6.95 ± 0.80 Titration range: 0.1–95.5 µM ΔA_max_ = 0.0099 type I

Lines of different colors indicate spectra at different ligand concentrations.

## Data Availability

The data presented in this study are available on request from the corresponding author. The data are not publicly available due to the IBOCH regulations.

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
