# Peer review of "Molecular Cloning, Heterologous Expression, Purification, and Evaluation of Protein–Ligand Interactions of CYP51 of Candida krusei Azole-Resistant Fungal Strain"

_biomedicines, 2023, doi:10.3390/biomedicines11112873_

Round 1
Reviewer 1 Report
I have to admit that I really liked this work. First of all, it was concise and nicely written but at the same time sufficiently informative. The aim was clearly stated, which was molecular cloning, purification and evaluation of the ligand-binding properties of C. krusei CYP51. I found this aim not only important but also successfully completed by the Authors. Therefore, I recommend acceptance of this work after some corrections, listed below.
Line 42, please list the most important Candida species here
Line 51, what are the current “last resource” solutions to treat those multiresistant species?
Line 52, please replace “drugs” with “active pharmaceutical ingredients”
The general structure of azoles should be introduced in the Introduction section, in the form of a figure.
Line 76, what is “и”?
Line 91, the structure of C. krusei CYP51 has been modelled before using i.e. homology models, i.e. look here: 10.1128/AAC.47.4.1213-1219.2003 . The Authors should have compared the structures obtained before with the current one.
Line 93, more details on the structure optimization must be provided
The study could be supported by the molecular docking and molecular dynamics analysis. The first one would enable to study the interaction between the chosen ligands and CYP51. There are now very accurate software, unfortunately mostly the commercial one, that allow the reliably determine the binding site within the protein. I would recommend using one of them. Also, the MD simulations would allow to check the dynamic stability of such complexes and the structure itself.
Table 4, some of those compounds (ligands) presented in Table 2 are really large. I strongly recommend some basic, quick and freely accessible serves, i.e. SwissAdme (http://www.swissadme.ch/) to predict the draggability of those molecules. I’m particularly worried about the Ch-4.
Reviewer 2 Report
The manuscript delivers interesting outcomes; however, there are some important points that need consideration.
1. The introduction section needs improvement, especially on the role of natural inhibitors alone or in combination with standard antifungals against Candida spp.. I recommend the authors use the reference (DOI: 10.1002/ptr.5650) to discuss this information.
2. Please provide a detail description (in the Materials and Methods section) of protein-ligand interaction. For example, how did you perform the molecular interaction? what kind of software was used? operational and setting conditions, and so on.
3. Please provide adequate references to all methods used in your study.
4. The results need to be more discussed and rationalized with previously published studies
5. This study lacks statistical analysis. Please clarify.
6. Please check that all scientific names are italicized.
The English usage requires moderate refinement. Therefore, I recommend the authors double-check the full text for grammatical and typing errors and seek a help of native English speaker.
Reviewer 3 Report
Authors proposed a paper entitled “Molecular Cloning, Heterologous Expression, Purification, and Evaluation of Protein-Ligand Interactions of CYP51 of Candida krusei Azole-Resistant Fungal Strain” for the publication in Biomedicines, MDPI.
This paper has a good scientific soundness and deserves to be published after major revisions.
Lines 36-38. I would include additional comments such as: “The widespread use of antifungal agents, both in clinical settings and agriculture, has exerted selective pressure on fungal populations, facilitating the development of drug resistance. Additionally, factors such as immunocompromised patients, invasive medical procedures, and global travel contribute to the rising incidence of fungal infections.”
Line 60-61. The definition of the aims and objectives of this paper should be expanded.
Line 105. Add a space among the previous and subsequent paragraph.
Line 145. “0.2% followed” on mass basis?
Line 164. Add space among paragraphs.
Line 190. Add space among paragraphs and try to link sub-paragraphs among them.
Line 230. I would rewrite the sentence as follows: A polished steel target containing 96 wells was employed for mass spectrometry analysis.
Line 274. “We also carried out a 3D …” I generally recommend to use impersonal forms.
Figure 6. The values of the peaks report decimals using comma and not points/dots. Maybe authors could approximate to unity.
Table 1. Please check the guidelines for this table.
Line 390. I would add some additional comments to this sentence, such as “This pressing issue not only underscores the importance of ongoing research and development in the field of antifungal medications but also highlights the significance of identifying novel drug targets and therapeutic approaches. In this context, interdisciplinary efforts involving molecular biology, genomics, and medicinal chemistry play a crucial role in the discovery of innovative antifungal agents that can combat drug-resistant strains, ultimately ensuring more effective treatments and improved outcomes for patients facing fungal infections.”
Table 2. Same observation of table 1.
Line 470. This sentence could be rewritten as “CYP51 enzymes found in pathogenic fungi serve as essential drug targets for the development of selective antifungal agents aimed at addressing azole-resistant fungal infections.”
Moderate revision of the use of English is highly suggested.
Round 2
Reviewer 2 Report
The manuscript has been sufficiently improved.
The English usage has been refined; however, minor revision is required during proofreading.
Reviewer 3 Report
Authors provided a revised version of their paper. Additional comments have been added in order to clarify the panorama in which authors are acting.
I requested to expand the paragraph in which authors define the goals of the work, in lines 90-91, at the end of the introduction section. Authors added 2-3 lines more; it can be sufficient.
Paper can be published now.